# Sepsis and Clinical Simulation: What Is New? (and Old)

**DOI:** 10.3390/jpm13101475

**Published:** 2023-10-08

**Authors:** Pablo Cuesta-Montero, Jose Navarro-Martínez, Melina Yedro, María Galiana-Ivars

**Affiliations:** 1Department of Clinical Simulation (SimIA Lab), Dr. Balmis General University Hospital, Alicante Institute for Health and Biomedical Research (ISABIAL), 03010 Alicante, Spain; pabloccuesta@yahoo.es (P.C.-M.); simia@isabial.es (M.Y.); 2Department of Anesthesiology and Surgical Critical Care, Dr. Balmis General University Hospital, Alicante Institute for Health and Biomedical Research (ISABIAL), 03010 Alicante, Spain; mgivars@gmail.com

**Keywords:** sepsis, clinical simulation, medical education

## Abstract

Background: Sepsis is a critical and potentially fatal condition affecting millions worldwide, necessitating early intervention for improved patient outcomes. In recent years, clinical simulation has emerged as a valuable tool for healthcare professionals to learn sepsis management skills and enhance them. Methods: This review aims to explore the use of clinical simulation in sepsis education and training, as well as its impact on how healthcare professionals acquire knowledge and skills. We conducted a thorough literature review to identify relevant studies, analyzing them to assess the effectiveness of simulation-based training, types of simulation methods employed, and their influence on patient outcomes. Results: Simulation-based training has proven effective in enhancing sepsis knowledge, skills, and confidence. Simulation modalities vary from low-fidelity exercises to high-fidelity patient simulations, conducted in diverse settings, including simulation centers, hospitals, and field environments. Importantly, simulation-based training has shown to improve patient outcomes, reducing mortality rates and hospital stays. Conclusion: In summary, clinical simulation is a powerful tool used for improving sepsis education and training, significantly impacting patient outcomes. This article emphasizes the importance of ongoing research in this field to further enhance patient care. The shift toward simulation-based training in healthcare provides a safe, controlled environment for professionals to acquire critical skills, fostering confidence and proficiency when caring for real sepsis patients.

## 1. Introduction

Sepsis stands out as a major public health threat, affecting millions of people worldwide and representing one of the leading causes of morbidity and mortality worldwide [1,2]. It is not a simple clinical condition; its complexity and severity make it a condition that requires immediate recognition and intervention to increase the patient’s chances of survival [3]. However, effectively addressing this challenge requires not only a theoretical understanding of the problem, but also specialized practical skills and abilities [4]. In recent decades, the use of simulation as a learning tool in healthcare education has boomed. It offers healthcare professionals a unique opportunity to hone their skills in managing clinical situations in a structured and controlled learning context [5,6]. This shift in medical pedagogy has changed the traditional dynamics of healthcare education and training, often based on the ‘see one, do one, teach one’ approach [7,8]. However, in today’s world, where patient safety is paramount and margins for error are increasingly narrow, the need for a risk-free learning environment for both patients and practitioners is increasingly evident and necessary [9,10]. As technology and educational methods continue to evolve, clinical simulation is no longer confined to simulation centers [11]; it is being integrated into hospitals and clinics during specific processes and situations, becoming ‘in situ’ and thus expanding its scope and relevance [12,13]. In the context of sepsis, this approach becomes even more important as it allows for the identification of problems in the workplace and the implementation of real changes [14,15,16,17]. Clinical simulation is a toolbox that ranges from simple exercises to very complex simulations, creating scenarios that aim to hone not only individual skills, but also communication and teamwork skills, with a common goal centered around preparing healthcare professionals for the real world [18,19]. This article aims to highlight the relevance of clinical simulation as a pedagogical innovation to address the medical challenges posed by sepsis, as well as the ability of this tool to generate change in healthcare professionals and improve adherence to clinical guidelines.

## 2. Sepsis, a Global Education Issue

Approximately 48.9 million cases of sepsis and 11 million sepsis-related deaths were reported worldwide in 2017, with mortality rates steadily increasing and an estimated annual cost of more than USD 24 billion in the US [20].

Of the global initiatives to address this major problem, the Surviving Sepsis Campaign (SSC) is undoubtedly the best known. The initial objectives of the campaign were to increase health, social, and political awareness of the disease; to improve early diagnosis through a clear definition of sepsis; to improve treatment; to promote education; to improve post-ICU care; and to implement global guidelines [21].

From the first edition in 2004 [22] to the latest edition in 2021 [3], these guidelines have had a global scientific impact with more than 20,000 citations and have been translated into more than 25 languages [23]. These guidelines revolve around essential care packages or bundles for sepsis management that aim to improve care, reduce mortality, and improve outcomes [24,25,26]. A bundle is a set of diagnostic and therapeutic interventions, based on clinical guidelines and supported by current evidence, that, when used together in an orderly fashion and over a period of time, have a greater benefit on the patient than when used in isolation [27].

Despite the proven effectiveness of these sepsis care bundles, adherence is suboptimal and sepsis is often under-diagnosed [28]. Educational interventions have been shown to improve bundle adherence in many studies, but are not always associated with improved patient management [29].

Most educational efforts have focused on ICU professionals, leaving a gap for other professionals who are critical in the early recognition of sepsis [30,31,32]. Preparing all healthcare professionals and students by improving understanding and knowledge of sepsis [2], as well as informing the general population and patients in particular, remains an unmet need [33].

Medical education interventions can be analyzed in different ways and with different tools [34], although the most interesting are those that measure their acceptability, the learning they provide, the impact they generate, and the duration of the change they produce [35,36]. Those related to sepsis are very diverse, with a structure and results that are difficult to extrapolate [37].

A recent review [32] found that the vast majority of educational initiatives were based on SCC bundles and their usefulness in early diagnosis and treatment algorithms, with less emphasis on other aspects (such as teamwork and communication).

Of the most commonly used educational strategies in medicine [38], traditional didactic teaching, e-learning, gamification, multimodal training, and clinical simulation appear in the literature [32,37].

Traditional didactic teaching is the oldest and most common method of approaching sepsis in the literature [39]. In this category, we find everything from seminars and sessions with slide presentations to articles and bibliographic material. This method is the simplest to organize and implement and is widely accepted as it follows the classical lines of education [38]. It has the disadvantage that knowledge retention can be limited compared to other methods that use active learning [40] and require physical spaces [41].

E-learning is one of the fastest growing forms of teaching at the moment, due to the advantage of being able to learn anywhere and anytime [42,43]. In recent years, and for obvious reasons, it has become one of the most widely used methods [41]. One of the evolutions of this system is blended learning, which combines face-to-face learning with e-learning and is gaining followers by combining the advantages of both systems. So much so that, in terms of educational benefits, it is an educational method in itself [44].

Gamification, or game-based learning, uses the attributes and dynamics of games to enhance the learning experience. These strategies are better accepted by professionals, especially younger ones, and show higher knowledge retention [45]. This methodology has been applied in various ways, from the realization of platforms similar to classic score-based video games [46] and simulation games [47] to the realization of escape rooms [48].

Multimodal training aims to cover all training spectrums, combining the best of each. Although some studies have demonstrated their impact on training, they remain a minority training modality [49,50].

Although many studies report improvements in patient care and outcomes following the implementation of a sepsis education intervention, regardless of the type of intervention, it is important to recognize that there are other variables that influence outcomes [29]. Hospital protocols, the internal workings of a hospital, the prior training of professionals, interprofessional relationships, and the culture of safety and teamwork are key elements in implementing any change [51,52,53].

Clinical simulation has been identified in the literature as one of the learning modalities with the greatest impact on professionals and the highest retention of knowledge when conducted in an appropriate manner [54]. Despite its benefits, it is by far the most underutilized educational intervention.

In order to improve sepsis education, and in order for this education to have an impact on clinical outcomes, it is necessary to (a) make the general population aware that it is a public health problem, (b) use educational interventions that are integrated with health policies and that create synergies with hospital protocols with current scientific recommendations as a central working axis, (c) ensure that these interventions must be based on active learning with a focus on simulation techniques, (d) involve all actors involved in patient care, and (e) provide care in multidisciplinary teams to centralize sepsis care in a multidisciplinary and multiprofessional approach.

Addressing the global educational challenge that sepsis represents necessitates the deployment of impactful and encompassing educational interventions aimed at enhancing its early recognition and management. A spectra of educational strategies, ranging from traditional didactic teachings to clinical simulation, are pivotal to counteract this medical condition; however, adherence and implementation are still found lacking in several instances. The educational endeavors should be holistic and multifaceted, and must promote active learner engagement, with an emphasis on clinical simulation, to ensure optimal knowledge retention and clinical application. Moreover, sepsis education should not only target healthcare professionals but also the general populace, raising awareness about this severe public health issue. Ultimately, the realization of educational interventions should align seamlessly with health policies and hospital protocols, ensuring a maximized impact on the clinical outcomes of sepsis patients

## 3. Clinical Simulation: A Toolbox

In order for an initiative such as the SSC to achieve the objectives for which it was created, it is essential to understand the barriers to its implementation [37,55]. It is logical to think that in order to tackle such a broad problem, it is necessary to use versatile, dynamic tools that can be integrated into all the processes and structures of healthcare systems, such as clinical simulation [56].

Clinical simulation is a technique, not a technology [56], that provides professionals with a set of tools through the use of controlled and safe practical experiences [57]. The methodology offers healthcare professionals, students, and participants the opportunity to acquire and refine clinical skills [6], make decisions [18] and experience complex situations without compromising their own safety or that of the patient [58].

As well as being used to improve practitioners’ skills and knowledge, clinical simulation provides an environment conducive to the study of various aspects of medical care, from the effectiveness of new interventions to human factors in healthcare [59,60]. By enabling data collection in a controlled environment, simulation becomes a valuable tool for clinical research that can lead to improvements in healthcare practice and policy [10].

Clinical simulation is particularly useful for training multidisciplinary teams to manage complex situations [19]. By simulating a clinical environment, critical elements of communication, teamwork, and decision making can be addressed, as well as aspects that are fundamental to patient care but often overlooked in other educational approaches [61].

This educational methodology has the capacity to adapt to different levels, healthcare disciplines, and healthcare settings, as well as to the different skills, technical or non-technical, that need to be taught [56]. This adaptability allows resources to be managed according to educational needs in order to optimize time and resources [62,63].

The development of educational activity through clinical simulation usually follows a structured educational model that at least includes the following phases [64,65,66] (Figure 1):

*-The identification of learning needs and the development of learning objectives*. Needs are defined as the actual gap between an individual’s practical performance and the health system’s intended performance for that position. This process can be as simple as reviewing the curriculum needs for a particular specialty or process across specialties [67], or it can be quite complex, involving in-depth analysis and more specific methodologies [68]. Sometimes, it is the simulation itself that is used to identify needs that cannot be diagnosed by other methods [69].

*-Planning*. The organization of available resources by teachers and learners. In this part, it is necessary to be very realistic about the objectives and the resources available. All the details of the organization of this type of event must be taken into account, from the timetable to the physical facilities to the teacher–student ratio (depending on the activity to be carried out) [70].

*-Prebriefing.* This refers to the information and orientation provided to participants before a simulation activity is carried out based on its scenario, objectives, and development [71]. This information mainly includes the following points [72,73]: the psychological safety of the environment, the learning expectations of the trainer, the time available for the activity, the simulation scenario, the working environment and logistics, and the description of the debriefing afterwards. It should be noted that the most important point is that of psychological safety, where rules of communication, confidentiality, and respect are established, as well as a fictional contract to accept the simulation as real [59].

*-Simulation*. A scenario should create the context in which the simulation is carried out according to the objectives that are to be achieved. In the field of clinical simulation, the structure of this scenario and the resources used, together with the experience of the facilitators, favor the realism of the simulation and improve the acquisition of knowledge and skills. Given that several elements interact during clinical simulation, it is advisable to use checklists or templates that improve and standardize the simulation and allow for subsequent evaluation [74,75,76].

*-Debriefing.* This last point could be defined as a learning conversation between several people to review a real or simulated event, analyzing their actions and emotions. During this process, guided reflection takes place in order to improve or maintain their performance in the future. This process aims to promote the development of clinical judgement and critical thinking skills [71,77]. Debriefing is recognized as the most important point within a clinical simulation activity [64,78,79,80]. Multiple structures and forms of debriefing exist in the literature [80,81], and although debriefing with good judgement is currently preferred [77], it is essential to adapt it to a specific educational process [63], and, above all, to conduct it [82]. One of the benefits of debriefing is its use in clinical practice, with the aim of reflecting on, and learning from, a real clinical situation [83,84,85]. This tool allows us to identify opportunities for improvement through direct observation, identify team needs, and create safe learning environments in the workplace [86].

Like clinical debriefing, clinical simulation should be integrated into the day-to-day running of care systems to compensate for the lack of real experience or to improve skills that are now fundamental to human treatment [56]. The translation of simulation into real life has benefits beyond those associated with learning alone, which are difficult to assess in the short term [87].

In situ simulation is an activity that takes place in the normal workplace rather than in simulation centers. It is a strategy based on creating scenarios, that are safe for the practitioner and the patient, in the real clinical environment, and it is used to identify the skills, knowledge, and teamwork required to develop appropriate clinical care. The applications of in situ simulation are very diverse, both clinical and educational, as well as assessment and quality [13,88].

New technologies, such as virtual reality, have been integrated into clinical simulation to improve participant acceptance and outcomes [89]. However, the most important role in clinical simulation is not played by the technology, but rather by the instructor/facilitator who is responsible for developing the simulation activity [90].

Far from the horizon described by David Gaba in 2004, simulation has spread, albeit unevenly, across the medical education landscape. The cost of implementing simulation clinically depends on the target population, the purpose of the simulation, and the technology required to implement it. It also depends on how educational and clinical organizations manage to reorganize their working structures in order to incorporate simulation.

Quantifying the benefits of simulation is much more difficult than assessing the costs. Most of the benefits, in terms of patient safety, error reduction, and improved performance and quality of care, are long-term and can only be realized through the consistent and prolonged use of simulation [56].

Despite the costs, the versatility of this “educational toolbox” makes it an essential strategy for preparing future health professionals, although, as with any educational method, its effectiveness depends on its correct use [65].

As healthcare continually evolves, the role of clinical simulation becomes even more pivotal in bridging the gap between theoretical knowledge and real-world applications. The dynamic nature of simulation ensures that healthcare professionals are not only equipped with knowledge, but also with the adaptability to navigate unforeseen challenges. Future projections suggest an increased reliance on simulation to address emerging health threats and to prepare for scenarios that traditional training cannot replicate. The investment in simulation, both in terms of time and resources, is an investment in patient safety, care quality, and overall healthcare advancement. With technology integrating further into healthcare, the fusion of advanced tech and simulation promises a transformative era of medical training and patient care.

## 4. From Theory to Practice

A recent systematic review of sepsis education [32] found that simulation-based learning models improved knowledge acquisition and retention and teamwork skills compared to standard teaching methods, developing fundamental skills for the early management of sepsis such as critical thinking. One of the findings of this study is that clinical simulation represented the minority of educational practices used for teaching sepsis.

Despite the importance of the sepsis problem and the benefits of clinical simulation, there are few high-quality studies in the literature that reflect its use [32,91]. However, as with evidence-based medicine, it is necessary to integrate individual practical experiences with the best available evidence to provide the best possible care [92], or, in this case, the best possible education.

Changes in aviation safety or in the practice of anesthesiology were not based on evidence that certain practices reduced the incidence of complications, but on the widespread implementation of small changes, of which few were subjected to controlled experiments [93]. As in these disciplines, the benefits of simulation cannot always be measured by controlled experiments [94], nor can they be measured with the usual tools in terms of educational impact [35] or quality [95].

However, the current system of education, training, and competence maintenance has not been subjected to rigorous testing to determine whether it actually meets its stated objectives [56].

The most frequent purpose for which this tool is used is the assessment of knowledge and competencies [14,96], most often within multi-modal training [32]. There are many advantages of this tool, as it allows for the assessment of knowledge between different specialties [97] and competences within the same specialty with different degrees of experience [98]. Another purpose for which it has been used is for training in an initial approach to recognize the septic patient [99].

One of the most important issues worked on through simulation is the effect of interprofessional working relationships during the treatment of sepsis [51,100,101]. Despite appearing as a fundamental point in the treatment of time-dependent pathologies [102,103], and in particular in sepsis [3], it is one of the points to be improved in simulation training [32,91].

Simulation can be used as a tool to train all healthcare professionals who are in contact with the patient, regardless of their specialty and category. Nursing plays a key role in this [32,37] and simulation can provide a safe environment to improve skills, knowledge, and care [104].

Knowledge of and training in sepsis should not be limited to healthcare professionals [33]. Staff-in-training programs have been shown to be feasible and effective [105] in improving diagnostic skills in multiple clinical settings and across multiple specialties [101,106,107,108,109,110].

The incorporation of simulation into the training curricula of medical and nursing students has produced very good results in terms of both knowledge and skills in the diagnosis and deterioration of patients with sepsis [111,112].

Outside the hospital, and for less skilled and less experienced healthcare workers, simulation is of considerable benefit in managing rare events and identifying sepsis-related problems [113].

Beyond recreating complex scenarios in intensive care units, simulation can be applied at all points of care of the septic patient. From diagnosis in long-term care homes [113], admission to the emergency department, passage through intermediate care units [114], the hospital ward [115], or the operating theatre [109], clinical simulation can replicate the situations in which patient care takes place. The development of in situ simulation enables training in complex settings where the early diagnosis and treatment of sepsis is critical, such as emergency departments [16] and rural hospitals and health services [115,116].

Sepsis management requires both technical and non-technical skills, all of which benefit from simulation, not only for the learning itself but also for patient safety. The most common example is central line cannulation [117]. Non-technical skills, probably more important and less studied, comprise another fundamental area in sepsis management, allowing for improved functioning and workload during teamwork [118].

The type of skills to be trained in sepsis management will determine the technology required for simulations, bearing in mind that technology and fidelity are not the most important elements in a simulation [119,120]. Virtual simulation, telesimulation, and augmented reality are very new tools that can help improve learning experiences in sepsis, although they are currently underdeveloped in the field [121,122,123].

Post-simulation debriefing aims to promote change in clinical practice through reflection [78]. Although reflective debriefing is probably the most widely used to reinforce knowledge in sepsis simulation scenarios [124], other more recent simulation methods, such as Deliberate Rapid Cycling Practice [125], have been shown to be equally effective and even superior in the post-training period for the management of septic shock [54].

Clinical debriefing following the clinical deterioration of patients has been shown to be a powerful educational tool [126], especially when used in a multidisciplinary manner [127], with multiple benefits for healthcare workers [128].

In some of the studies mentioned above, the underlying evidence provided strong support for the benefits of simulation. In many cases, however, both the authors and the institutions involved in these studies proceeded with a limited evidence base because of the superficial consistency of the simulation approaches employed. The difficulty of conducting definitive research on their success, together with continuing dissatisfaction with traditional methods, also influenced this decision.

Simulation is thus not merely a replicative tool but a transformative approach to medical education and patient care, necessitating a paradigm shift in our perception and utilization of educational methodologies in healthcare. The multifaceted applications of simulation underscore its instrumental role in enhancing interdisciplinary collaboration and optimizing patient outcomes in sepsis management. As we navigate the complexities of healthcare, the incorporation of innovative simulation techniques will be pivotal in fostering an environment of continuous learning and improvement. The progressive integration of simulation in medical education heralds a new era in healthcare, characterized by enhanced clinical proficiency and improved patient care paradigms.

## 5. Conclusions

Clinical simulation has transformed the way professionals prepare for clinical situations, replacing the traditional approach with a structured and controlled methodology. This technique allows skills and competencies to be honed in a safe environment, prioritizing patient safety and minimizing medical errors. Its accessibility has expanded beyond simulation centers into real clinical settings, broadening its scope and relevance.

In the context of sepsis, clinical simulation becomes even more important as it allows for problem identification and resolution in real working environments. In addition to improving individual skills, it promotes teamwork and communication skills, preparing professionals to face real-world challenges. Adherence to key clinical guidelines and care bundles is critical in the management of sepsis, and education plays a critical role in improving this adherence.

Simulation is adaptable to a variety of educational strategies, from traditional teaching to gamification and multimodal training. Gamification, for example, has been shown to be particularly effective with young professionals, increasing knowledge retention. However, factors such as hospital protocols, the prior training of professionals, and the culture of safety and teamwork also influence outcomes.

In conclusion, clinical simulation is an essential tool in addressing the challenges of sepsis. Its ability to enhance skills, promote collaboration, and improve adherence to clinical guidelines makes it a valuable strategy in the training of healthcare professionals. The pedagogical innovation that it represents has the power to bring about real change in professionals, thereby improving care and patient safety in the fight against sepsis. Clinical simulation can be adapted to different stages of patient care, in different clinical settings and for different levels of experience, demonstrating its versatility and effectiveness in sepsis management training.

## Figures and Tables

**Figure 1 jpm-13-01475-f001:**
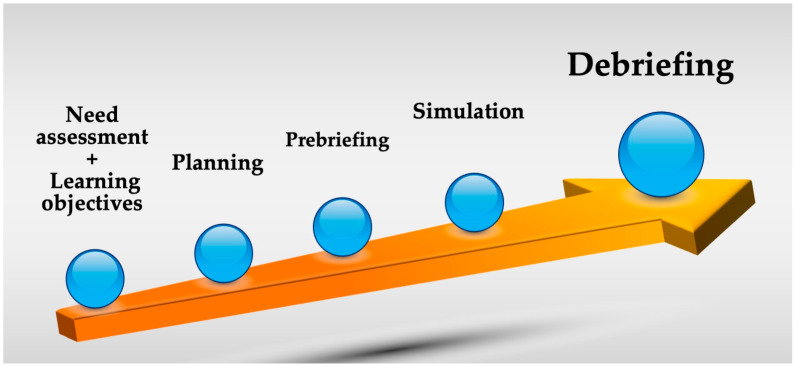
Phases of a structured educational model in clinical simulation.

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
