# Peer review of "Sepsis and Clinical Simulation: What Is New? (and Old)"

_jpm, 2023, doi:10.3390/jpm13101475_

Round 1

Reviewer 1 Report

The review offers a comprehensive understanding of the sepsis education issue. The manuscript is very well-written, its structure is easy to navigate and its title and abstract reflect the content of the manuscript. I genuinely enjoyed reading the manuscript and found it very educational and evidence-based. I haven't detected any significant issues with the manuscript and recommend it for publication as is. I'm confident that it will be very educational for the Journal's audience and well-received.

Author Response

Thank you very much for your comments.  Our aim was to produce a really useful and practical article.

Reviewer 2 Report

In this work the Authors discuss the importance of simulating clinical scenarios about sepsis/septic shock. Since sepsis is one of the disease with the highest mortality rate, I strongly agree with the Authors in highlighting the importance of teaching treatment strategies with simulation.

Overall, this manuscript is well-designed and provides different interesting aspects. Anyway, there are a few concerns that should be addressed:

1.     The Abstract is far too long and unstructured. Authors’ instructions state that “The abstract should be a total of about 200 words maximum. The abstract should be a single paragraph and should follow the style of structured abstracts, but without headings: 1) Background: Place the question addressed in a broad context and highlight the purpose of the study; 2) Methods: Describe briefly the main methods or treatments applied. Include any relevant preregistration numbers, and species and strains of any animals used; 3) Results: Summarize the article's main findings; and 4) Conclusion: Indicate the main conclusions or interpretations”. Please provide this changes.

2.     Introduction: the Authors stated “It is not a simple clinical condition; its complexity and severity make it a condition that requires immediate recognition and intervention to increase the patient's chances of survival”. I agree with this concept. However, I would expand it just a little highlighting the importance of a multilateral treatment strategy. In this sense, I suggest the Authors to cite one of the most recent review about this topic, i.e. Guarino M, Perna B, Cesaro AE, Maritati M, Spampinato MD, Contini C, De Giorgio R. 2023 Update on Sepsis and Septic Shock in Adult Patients: Management in the Emergency Department. J Clin Med. 2023 Apr 28;12(9):3188.

3.     Clinical simulation, lines 163-201. I found this part really amazing and it clearly expressed the main parts of what each clinical intervention in emergency setting should provide. I humbly suggest to create a small table to summarize the main concept of this paragraph.

Minor revisions are advisable

Author Response

Thank you very much for your review.

We have changed the format of the abstract to make it simpler (adding a graphical abstract) and included the reference. 

Clinical simulation, lines 163-201. We have increased the spacing between key points to make them more visual.